# Definition of the Surgical Case Complexity in the Treatment of Soft Tissue Tumors of the Extremities and Trunk

**DOI:** 10.3390/cancers14061559

**Published:** 2022-03-18

**Authors:** Annika Frei, Mario F. Scaglioni, Pietro Giovanoli, Stefan Breitenstein, Philip Heesen, Bruno Fuchs

**Affiliations:** 1Institut für Pathologie, Kantonsspital Winterthur, 8400 Winterthur, Switzerland; 2Klinik für Hand- und Plastische Chirurgie, Luzerner Kantonsspital, 6000 Lucerne, Switzerland; mario.scaglioni@luks.ch; 3Klinik für Plastische Chirurgie und Handchirurgie, Universitäts Spital Zürich, 8091 Zurich, Switzerland; pietro.giovanoli@usz.ch (P.G.); fuchs@sarcoma.surgery (B.F.); 4Departement Chirurgie, Kantonsspital Winterthur, 8400 Winterthur, Switzerland; stefan.breitenstein@ksw.ch; 5Medizinische Fakultät, Universität Zürich, 8032 Zurich, Switzerland; philip.heesen@uzh.ch; 6Klinik für Orthopädie und Unfallchirurgie, Luzerner Kantonsspital, 6000 Lucerne, Switzerland; 7Klinik für Orthopädie und Traumatologie, Kantonsspital Winterthur, 8400 Winterthur, Switzerland

**Keywords:** soft tissue tumors, complexity score, sarcoma

## Abstract

**Simple Summary:**

Soft tissue tumors are heterogeneous tumor entities that often require surgical intervention for treatment. While some tumors are easy to resect, others require extremely complex, interdisciplinary surgery depending on the tumor type, localization and biological behavior. Up to now, there has not been an instrument able to objectify the complexity of such a surgery; therefore, we attempted to establish a complexity score for the description of soft tissue tumor surgeries. Furthermore, we aimed to categorize surgeries in such a way that patients can be assigned the best treatment such that a cost-effective approach can be taken.

**Abstract:**

Background: We intend to establish a complexity score for soft tissue tumor surgeries to compare the complexities of different soft tissue tumor surgeries and to ultimately assign affected patients to appropriate treatments. Methods: We developed a soft tissue tumor complexity score (STS-SCS) based on three pillars: in addition to patient-related factors, tumor biology and surgery-associated parameters were taken into account. The STS-SCS was applied to our sampling group of 711 patients. Results: The minimum STS-SCS was 4, the maximum score was 34 and the average score 11.4 ± 5.9. The scores of patients with malignant diagnoses were notably higher and more widely scattered than those of patients with benign or intermediate malignant tumors. To better categorize the complexities of individual surgeries, we established four categories using the collected data as a reference dataset. Each of the categories contained approximately one-quarter of the registered patients. Discussion: The STS-SCS allows soft tissue tumor surgeries to be retrospectively evaluated for their complexity and forms the basis for the creation of a prospective concept to provide patients with the right intervention in the right geographic location, which can lead to better results and provision of the most cost-effective overall treatment.

## 1. Introduction

Soft tissue tumors are rare, and affected patients often initially present to general practitioners or orthopedic surgeons [1]. The clinical differentiation between benign and malignant lesions is often very difficult, and even highly malignant soft tissue tumors are often misdiagnosed as benign tumors [2]. A reliable diagnosis can often only be made by biopsy, which is the only way to determine the histological subtype and grade according to the FNCLCC system [2]. Unfortunately, soft tissue sarcomas are often not primarily recognized as such, so the term “whoops procedure” describes the situation where a lump is incompletely removed by a surgeon who is not aware of the malignancy of the soft tissue tumor. In this case, extensive subsequent re-excisions are often required as residual tumor tissue is a risk factor for local recurrence [3].

Surgery is the mainstay of therapy for both benign and malignant soft tissue tumors. Sarcomas originate from the entire skeleton and surrounding soft tissues and display various biological behaviors that are dependent on the biologic entity. The goal of surgery is complete en bloc resection with avoidance of positive margins, whenever possible, to reduce the risk of local recurrences, distant metastases and mortality [4]. Surgical techniques of resection are various and often depend on the anatomic site of the lesion. Furthermore, while some resections are followed by complex reconstructions, others require no further surgical interventions. In addition, chemotherapy and radiotherapy are considered important pillars of multimodal and transdisciplinary sarcoma treatment, either preoperatively, postoperatively or in combination. All of these aspects are evidence that surgery for soft tissue sarcomas is a highly complex transdisciplinary action that needs to be personalized for each patient’s specific situation.

There is much debate regarding centralization of complex surgery to reduce costs; however, there are no robust data to define surgical complexity, which is obviously a critical determinant because there is a wide spectrum for a given disease. Defining the complexity of surgical resection in terms of center-based medicine is important for several reasons. As early as 1979, Luft et al. postulated a correlation between the surgical volume and mortality. They showed that for various complex interventions, mortality seemed to be inversely proportional to the volume of operations [5], which was also confirmed by others [6]. Despite a wealth of data, recent studies have highlighted various challenges facing centralization efforts [7]. Volume-based morbidity improvements do not seem to be transferrable to all surgeries, with some studies concluding the opposite [8]. A high surgical volume does not guarantee a good outcome for all types of surgeries, and poor processes may become naturalized in centers due to frequent repetition [7].

Modern healthcare concepts, therefore, include integrating the complexity of a procedure and the complexity of a patient (with associated comorbidities) to determine the optimal location for care [9]. With the advent of value-based healthcare delivery, the definitions of quality and outcomes are pivotal to defining the value for the patient, in addition to the cost package over the full care cycle. In most hospitals, costs are still defined by diagnosis/volume-based accounting systems, which by no means reflect the complexity of soft tissue sarcoma surgery. For all these reasons, we aimed to establish a score for the complexity of soft tissue tumor surgery to enable comparison within a diverse surgical spectrum. As sarcomas are rare, occur in all anatomic locations of the body and their treatment is highly multidisciplinary, surgical treatment involves a wide spectrum of complexity, includes both resections and reconstructions and may, therefore, ideally be suited to such an analysis of surgical complexity.

## 2. Materials and Methods

### 2.1. Study Population

All surgeries on soft tissue tumors over a 15-year period performed by a single surgeon were registered in the AdjumedCollect “Sarcoma Surgeon’s Registry” (Adjumed Services AG, Zurich, Switzerland; www.adjumed.ch (accessed on 22 October 2021)). The AdjumedAnalyze tool (Adjumed Services AG, Zurich, Switzerland) can be used for basic statistics, such as combinations of parameters, and allows the extraction of data. The individual scores were subsequently calculated in Microsoft Excel (Microsoft Corporation, Redmond, WA, USA).

### 2.2. Soft Tissue Tumor Surgery Complexity Score (STS-SCS)

Based on the literature and expertise of experienced sarcoma surgeons, we compiled and defined relevant parameters for an STS-SCS. The score is essentially based on three pillars: the patient, tumor biology and surgery-based parameters (Figure 1).

In the first pillar, patient-related factors such as age and prior history—in particular, previous radio- or chemotherapy—were summarized. It was shown that elderly patients with soft tissue sarcomas of the extremities have lower overall survival compared with younger patients [10]. Neoadjuvant therapies such as radio- or chemotherapy play an increasing role in the treatment of soft tissue tumors and must be included in perioperative management. The number of available neoadjuvant modalities is constantly increasing, and the use of possible therapies must be individually assessed due to the heterogeneity of tumors. Patients with advanced disease, in particular, can often benefit from neoadjuvant therapy modalities, but there are also associated risks. For example, neoadjuvant radiotherapies may lead to wound-healing disorders, while chemotherapies can lead to a delay in surgical treatment or even tumor progression during chemotherapy. In addition, complications during neoadjuvant CHT can significantly delay surgery and prolong the overall treatment time [11]. Previous whoops operations for misjudged soft tissue tumors also complicate perioperative management and often entail excessive re-excisions to reduce the risk of local recurrence [3].

For patients with soft tissue tumors, it has been shown that the smaller the tumor at diagnosis, the better the prognosis [12]. Histological grading seems to be the most important factor for the prognoses of patients with soft tissue tumors and, thus, has an even higher significance than histological typing [13]. In addition to the histological type, the anatomical location of soft tissue tumors also seems to be decisive. For example, it has been shown that metastases occur more frequently in patients with sarcomas of the lower extremities than the upper extremities, and that these tumors are, by far, more frequently larger and deeper [14]. The centralization of soft tissue tumor surgeries in high-volume hospitals can especially improve the survival of patients with non-low-grade and deep-seated tumors [15].

Due to the mesenchymal origin of sarcomas, these tumors often involve multiple anatomical structures and regions [16]. Depending on the location and proximity to surrounding organs, the removal of several structures may be necessary for sarcoma resection. If only soft tissues were removed, a score of 1 was assigned; if other structures such as muscles, nerves, bones, periosteum, tendons or vessels had to be removed, additional points were given. This challenge may require the expertise of different surgical subspecialties, so a multidisciplinary treatment team will include surgical oncologists from several different specialties such as orthopedics, thoracic surgery, general surgery, vascular surgery, neurosurgery, urology and gynecology as well as reconstructive plastic surgery. In addition to the interdisciplinary challenges, the involvement of vascular structures also seems to have an influence on the recurrence rate. It was shown that when the tumor was radiologically surrounded by large vessels, vascular resection and bypass reconstruction provided improved local control [17]. The involvement of vascular structures in a sarcoma significantly complicates surgery but is not, in itself, a contraindication for sarcoma resection [18].

Following sarcoma resection, reconstruction is often not necessary; however, in selected cases, patients may benefit greatly. Reconstructive procedures can be, at times, extremely elaborate, depending on the type and extent. The heterogeneity in tumor reconstruction was thus taken into account with a procedure-specific evaluation system. However, it is obvious that not all possible types of reconstruction can be adequately represented by a score, especially since the possibilities are very extensive and the indication for each patient is individual.

Finally, the previously identified relevant factors were examined for their individual influences on the complexity of surgery and weighted accordingly. This resulted in a total score (Table 1) based on the listed parameters and their corresponding weighting, which was then individually determined for each patient, using the data extracted from Adjumed, by adding the individual factors together.

## 3. Results

### 3.1. Characteristics of Soft Tissue Tumor Patients

In this study, we examined the data of 711 patients. The mean age of the analyzed patients was 51.0 ± 18.2 years. Of the operated patients, 70% were between 18 and 64 years old. Males accounted for 383 of the patients, and 328 were females (Figure 2). The male-to-female ratio was 1.17. Of the patients, 263 had benign tumors (37%), 118 patients had tumors with intermediate malignancy (17%) and 270 patients suffered from malignant (38%) soft tissue tumors. The remainder of the patients (8%) had metastases, hematologic solid tumors or tumor simulators (a tumor that may imply a sarcoma on imaging but turns out to be a benign mesenchymal non-tumorous lesion). The most common benign diagnosis was, by far, lipoma (131 patients; 18%). We found that 51 patients (7%) had atypical lipomatous tumors, which are classified as tumors of intermediate malignancy. The most common malignant diagnosis was undifferentiated/unclassified pleomorphic sarcoma (UPS) (76 patients; 11%), followed by myxoid liposarcoma (42 patients; 6%) and myxofibrosarcoma (33 patients; 5%). Other diagnoses were much rarer.

### 3.2. Application of the STS-SCS

The STS-SCS was applied to our sampling group of 711 patients and the individual scores were calculated for each patient using Microsoft Excel. The minimum score was 4 and the maximum score 34, with an average score of 11.4 ± 5.9. The scores of patients with malignant diagnoses (17.5 ± 4.6) were notably higher and more widely scattered than those of patients with benign (6.8 ± 1.8) or intermediate malignant tumors (10.2 ± 4.1) (Figure 3).

### 3.3. Categorization of Soft Tissue Tumor Surgery Complexity

To better categorize the complexity of individual surgeries, we established four categories using the collected data as a reference dataset. Each of the categories contained approximately one-quarter of the registered patients (Table 2).

Category 1 included patients with a score lower than 7 points. This covers a relatively wide range of scores and included 180 (25.3%) patients. This category contained patients with benign, intermediate malignant tumors and tumor simulators. Patients with a score of between 7 and 9 points were assigned to category 2, which covered only a very small range of points, but still included 157 patients (22.1%). Most patients in this range had benign or intermediate malignant diagnoses, but there were also a few malignant diagnoses. Category 3 included patients with a score of 10–15 points and comprised 191 patients (26.9%). In the highest category, 4, there were 183 patients (25.7%) with almost exclusively malignant diagnoses, with a few exceptions. The highest category covered the largest range, and the scores were further apart from each other than those in the lower categories (Figure 4).

## 4. Discussion

In our study, we defined the STS-SCS based on three main pillars: patient-related factors (such as age and prior history), tumor biology (tumor size, histology/grading and anatomical location) and surgery-associated factors, such as the type of resection and reconstruction or the involved disciplines. This score was then applied to a sample group of 711 patients with various soft tissue tumor diagnoses. An individual score was calculated for each of the subjects. Based on these data, four complexity categories were defined that allow the assignment of the individual surgeries to four complexity levels. This strategy allowed the assignment of each soft tissue tumor operation to a complexity level and, thereby, the comparison of the different interventions.

This is the first time such an approach has been attempted; therefore, there were no alternative methods for comparison, which made the selection and weighting of factors challenging. It will never be possible to include all potentially relevant factors to fully describe a patient (e.g., comorbidities), but the current STS-SCS is intended to establish a basis for discussion. Just as particularly complex patients may not be adequately represented, unusually complicated surgeries had to be broken down and, therefore, may not yet be adequately covered by the selected categories.

In the value-based geography of care according to Porter et al. [9], the best possible cost-effective quality care for the patient is defined by the complexities of the procedure and the patient, which in an integrated system, allows the direction of patients requiring complex care to regional or central hubs, while those patients who need less complex care are moved to the most cost-effective local centers (Porter geography model). To realize such a model of geographically-based care, a tool to assess and define the complexity of a surgical procedure is mandatory; we have proposed such a tool, the STS-SCS, which maps the considerations into a complexity score. The integration of patient- and procedure-related factors allows the patient to be matched to the best possible treatment site as well as the optimal treatment team. Not only does the adequate allocation of patients increase the output, but it also allows the cost of treatment to be reduced for specific diseases or treatments [19]. Assigning the patient to the ideal treatment site seems to be an intuitive matter; however, its implementation, in practice, does not seem to be that easy, and the instruments for decision making are not yet available for sarcoma care. Therefore, the STS-SCS provides a tool to facilitate decisions related to the allocation of soft tissue tumor patients to appropriate treatment sites.

Considering complexity is not only important in terms of the treatment location but also the treatment team. Geography is a powerful tool to optimize value in three dimensions: the right mix of personnel, working together at the right location and with integration across time [9]. Porter et al. defined integrated practice units in which teams over a geographical region communicate and exchange to enhance the quality of care [9]. Patients with benign and malignant soft tissue tumors are often first seen by general practitioners or general and orthopedic surgeons, and making the correct diagnosis is often difficult, which frequently leads to unplanned excisions [20]. In addition, establishing the correct pathological diagnosis of a sarcoma is often difficult, and misdiagnoses often occur due to confusion with benign tumors [20]. Centralized pathological assessment of soft tissue tumors, for example, was shown to save costs while improving the quality of diagnoses [21]. Such an integrated exchange over a geographical region among multidisciplinary and cooperating integrated practice units helps to establish a complex diagnosis and initiate appropriate therapeutic measures [22]. The STS-SCS is an instrument that facilitates and objectifies the allocation of patients to the appropriate care site while considering their comorbidities and possible complications.

The definition of complexity for soft tissue tumor surgery using the STS-SCS also serves as a basis for assessing the quality of soft tissue sarcoma surgery. Up to now, it has been common practice to use the surgical volume as a predictor of the outcome [6], and for some soft tissue tumors, such as large, high-grade and retroperitoneal tumors, it has been shown that a good outcome is associated with a high volume [23,24]; similar results were also obtained for soft tissue tumors of the extremities [25]. Further to this, it has been shown that treatment in a multidisciplinary team improves the surgical margins for deep-seated lesions [26], while the French sarcoma group reported an impact on outcome by the multidisciplinary team approach, but interestingly not by surgical volume [27]. However, the definition of further quality indicators has been lacking until recently, which resulted in our intention to develop an approach to comprehensively assess the quality of sarcoma surgery. Certainly, and foremost, the quality of sarcoma surgery depends on the complexity of the procedure, which must be extensively considered when defining quality. Using the STS-SCS as a basis together with the extended database developed in the framework of this project, we can describe the complexity of a surgery as a common basis and, in a further step, use these tools to make considerations regarding quality.

The outcome for disease control and the quality of surgery not only depends on technical aspects but also on the correct diagnosis and, specifically, on the correct indication to perform the surgery [28]. Indication quality encompasses the appropriateness and necessity of medical interventions but continues to only be given a subordinate role in our current practice [28]. It is, therefore, crucial to establish scientific evidence and guidelines that facilitate the physician’s assessment of the appropriateness of an intervention. The STS-SCS greatly facilitates the ability to bundle specific procedures or groups of similar procedures for comparison and analysis and, thereafter, extrapolate to define the indication quality for performing a specific soft tissue tumor resection or reconstruction, thus making the indication quality an entry point for the quality discussion. Once the quality of a surgery is defined, this information can be extrapolated to the choice of the correct indication for the surgery regarding evidence-based principles and standards, which include the results of clinical studies’ and guidelines.

## 5. Conclusions

Currently, we are able to retrospectively assess surgeries according to their complexity using the STS-SCS, which was developed in our study. This score makes it possible, for the first time, to categorize soft tissue tumor surgeries based on their complexity, which allows patients to receive the right intervention or treatment at the right site, which may lead to better outcomes and more cost-effective treatment overall. Based on Porter’s principles and the STS-SCS presented in this article, a prospective approach to model soft tissue surgery evaluation was developed to assign patients with soft tissue tumors to the appropriate surgery site based on their individual risk factors and planned surgical intervention.

## Figures and Tables

**Figure 1 cancers-14-01559-f001:**
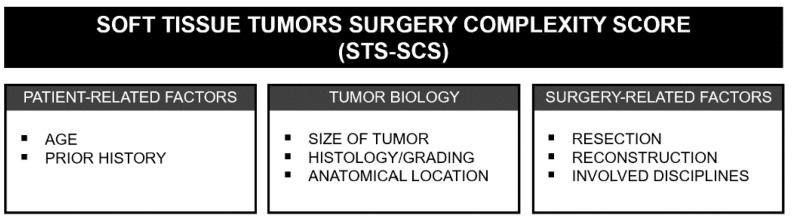
Overview of the three pillars and the individual factors on which the STS-SCS is based.

**Figure 2 cancers-14-01559-f002:**
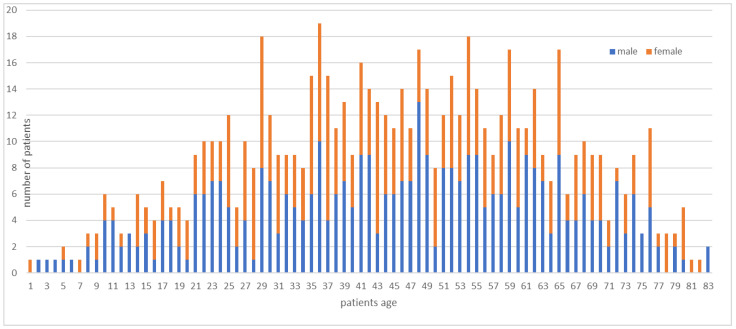
Age and gender distribution in the subject group.

**Figure 3 cancers-14-01559-f003:**
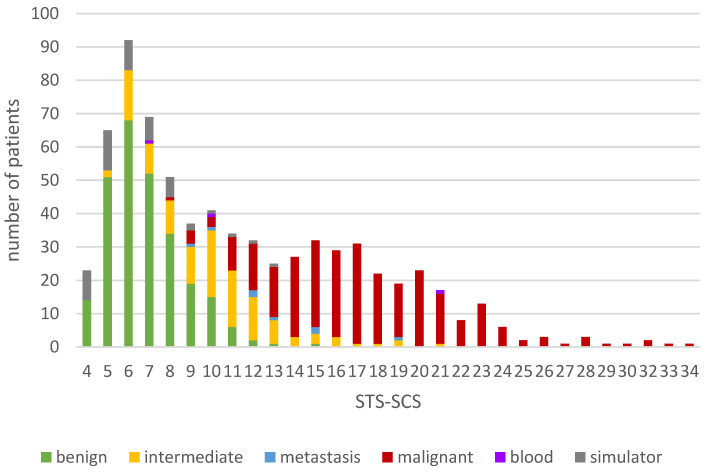
Distribution of the totals of the soft tissue tumor surgery complexity score (STS-SCS) in the sampling group.

**Figure 4 cancers-14-01559-f004:**
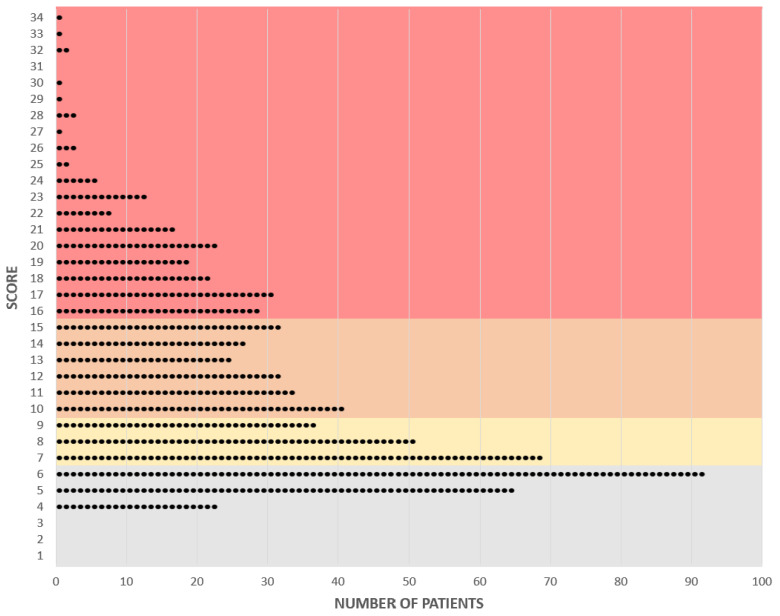
Graphical representation of individual scores and their allocation to the categories of complexity. Each point represents a patient. Category 1: gray, Category 2: yellow, Category 3: orange, Category 4: red.

**Table 1 cancers-14-01559-t001:** STS-SCS system indicating the weighting of each parameter.

			Points	Maximum
Patient’s Age	≤17 years		1	
	18–64 years		0	
	≥65 years		1	1
Histology/Grading	Benign		1	
	Simulator		1	
	Intermediate		2	
	Blood-based solid tumor		3	
	Metastasis		5	
	Malignant	G1	5	
	Malignant	G2	6	
	Malignant	G3	7	7
Prior History *	Preoperative radiotherapy		2	
	Preoperative chemotherapy		2	
	Prior whoops		2	6
Size of Lesion	5 cm or less		1	
	more than 5 cm, but no more than 10 cm		2	
	more than 10 cm, but no more than 15 cm		3	
	More than 15 cm		4	4
Anatomical	Superficial		1	
Location	Deep		2	2
Resected Structures (soft tissue, muscles, nerves, bones, periosteum, tendons, vessels) **	1		1	
	2		2	
	3		3	
	4		4	
	5		5	
	6 or more		6	6
Type of	Mesh graft		1	
Reconstruction ***	Tendon/ligament reconstruction		1	
	Bone cementation		1	
	Open reduction internal fixation (ORIF)		1	
	Bone autograft		2	
	Bone allograft chips		2	
	Other bone reconstruction		2	
	Vessel reconstruction		2	
	Nerve reconstruction		2	
	Lymphovenous reconstruction		2	
	Intra-abdominal reconstruction		2	
	Pedicled tissue transfer		3	
	Chest wall reconstruction		3	
	Free tissue transfer		4	16
Number of	One discipline		0	
Involved	Two disciplines		1	
Disciplines ****	Three disciplines		2	
	Four disciplines		3	
	Five and more disciplines		4	4
Total			max.	46

* The points in the section “prior history” can be added together, resulting in a maximum score of 6 in this field. ** For each resected structure (such as muscle, nerve, vessel, etc.) a point is added. *** The four highest scores in the section “type of reconstruction” are summed up. An intervention (for example, various nerve reconstructions) can be listed numerous times. **** If one single surgeon is sarcoma surgeon but has the credentials also for vascular reconstruction, then 2 disciplines are registered.

**Table 2 cancers-14-01559-t002:** Division of surgeries into four categories.

Category	Complexity Score	Number of Patients	Percentage (%)
1	≤6	180	25.3
2	7–9	157	22.1
3	10–15	191	26.9
4	≥16	183	25.7

## Data Availability

The data presented in this study are available on request from the corresponding author.

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
