# Peer review of "Definition of the Surgical Case Complexity in the Treatment of Soft Tissue Tumors of the Extremities and Trunk"

_cancers, 2022, doi:10.3390/cancers14061559_

Round 1
Reviewer 1 Report
Thank you for the opportunity to review this article. The article presents a classification system for the complexity level of management of soft tissue sarcomas. I think the topic is extremely interesting and the initiative potentially very useful. The article is well conducted and well written. I read this work with great interest. In my country, after a phase of particular centralisation of the management of these diseases, we are currently experiencing the reverse process, due to the increasing diffusion of surgical expertise needed for treatment. This is not necessarily a good trend. The use of a method allowing better stratification of the complexity of cases could optimise both the choice of treatment and especially the distribution of patients among different centres. I have no suggestions for the authors and I think the article is suitable for publication in its present form.
Author Response
Comment 1: Thank you for the opportunity to review this article. The article presents a classification system for the complexity level of management of soft tissue sarcomas. I think the topic is extremely interesting and the initiative potentially very useful. The article is well conducted and well written. I read this work with great interest. In my country, after a phase of particular centralisation of the management of these diseases, we are currently experiencing the reverse process, due to the increasing diffusion of surgical expertise needed for treatment. This is not necessarily a good trend. The use of a method allowing better stratification of the complexity of cases could optimise both the choice of treatment and especially the distribution of patients among different centres. I have no suggestions for the authors and I think the article is suitable for publication in its present form.
Response: We thank you very much for your encouraging words and are confident that our work can lay an important foundation for further considerations regarding center-based medicine in this field.
Reviewer 2 Report
The authors present three new groupings of prognostic factors for soft-tissue sarcomas.
This study is very interesting and reliable due to the large number of patients.
However, I would like to indicate some concerns that should be discussed.
- Why did they divide the patients into these three groups? Please indicate your rationale.
- All of these grouped prognostic factors have been reported in the past. Therefore, is it possible to add new prognostic factors?
- I do not think that benign and intermediate group tumors should be lumped together. What do you think about that?
- Are these grouped prognostic factors the same indicators at any age? For example, is the same true for the elderly over 65 years of age or for adolescent and young adults? Please cite the following articles, if possible, in your response.
-Surgical management of sarcoma in adolescent and young adult patients. BMC Res Notes. 2020 May 26;13(1):257. doi: 10.1186/s13104-020-05107-0. PMID: 32456671; PMCID: PMC7249334.
- Clinical features and outcomes of primary bone and soft tissue sarcomas in adolescents and young adults. Mol Clin Oncol. 2020 Apr;12(4):358-364. doi: 10.3892/mco.2020.1994. Epub 2020 Feb 4. PMID: 32190320; PMCID: PMC7058051.
- Clinical outcomes of patients with primary malignant bone and soft tissue tumor aged 65 years or older. Exp Ther Med. 2019 Jan;17(1):888-894. doi: 10.3892/etm.2018.7013. Epub 2018 Nov 26. PMID: 30651877; PMCID: PMC6307412.
Author Response
Comment 1: Why did they divide the patients into these three groups? Please indicate your rationale.
Response: We are not quite sure which groups you are referring to in your question: Are you referring to the components or the categorization of the score? The reason for including patient- and surgery-related factors as well as tumour biology parameters is that these are certainly the most important indicators of complexity of surgery. However, the parameters of this score are not to be considered exclusive and can be extrapolated accordingly if necessary. Patients were then divided into 4 groups depending on the result of the total score (please refer to the description in the M&M). This allows patients to be assigned directly to the appropriate treatment sites according to Porter's geography model of care. We kindly ask you to clarify your question if we have not answered it adequately.
Comment 2: All of these grouped prognostic factors have been reported in the past. Therefore, is it possible to add new prognostic factors?
Response: We are continuously collecting data and are confident that this work will provide the basis to evaluate further prognostic factors in the future, which certainly then can and should be included. It is not the aim of this paper to evaluate prognostic factors or to make statements about output. The score is intended to provide an objective description of the complexity of surgeries.
Comment 3: I do not think that benign and intermediate group tumors should be lumped together. What do you think about that?
Response: Benign and intermediate malignant tumors were separated in the score in different areas. These dignities are not lumped together and are reflected in the score in the same way such as malignant or metastatic tumors. The intermediate malignancy of the tumours is not only reflected in the histology/grading category, but also in size and location, since atypical lipomatous tumours, for example, are often large (> 15 cm) and are characterized by their special location (e.g. retroperitoneum). We are therefore confident that the intermediate malignancy of these tumors should be well reflected in the score.
Comment 4: Are these grouped prognostic factors the same indicators at any age? For example, is the same true for the elderly over 65 years of age or for adolescent and young adults? Please cite the following articles, if possible, in your response.
Response: Age is also included as a parameter in the global score, as older or younger patients may represent more challenging circumstances, as reflected by the score. However, we herewith only try to establish an objective baseline for surgical complexity, encompassing all potential factors regarding complexity. Our score is not relevant to prognosis at this stage and does not highlight the outcome of patients, how this will be evaluated prognostically will be the subject of further studies. Thank you very much for your literature suggestions, we will gladly take them into account for further considerations regarding outcome prediction.
Round 2
Reviewer 2 Report
The authors answered well, so the manuscript is suitable for publication.
This manuscript is a resubmission of an earlier submission. The following is a list of the peer review reports and author responses from that submission.
Round 1
Reviewer 1 Report
In this study, you propose a novel score to grade the complexity of the treatment of patiens with resectable soft tissue tumors. While the concept per se does make sense, your approach has a number of limOther factors such as possible preoperative therapies (chemotherapy, radiotherapy) are subject to individitations. One of the main issues I have with the score is that many of its factors can only be ascertained postoperatively (histology, number of resected structures). This substantially decreases its significance as a tool for prospective treatment planning.
Moreover, the significance and meaning of the score as an absolute number is very unclear. To validate its clinical implications, it should be correlated to postoperative outcomes (e.g. morbidity and mortality as well as functional outcomes) and to cost of care. Without such an analysis, you create a rather arbitrary number for patients.
A number of further issues require improvement throughout the manuscript.
- The introduction is rather long and not sufficiently focuses. You touch a lot of topics (cost of health care, centralisation, value-based health care) which only indirectly or marginally have to do with the topic of your study. I therefore suggest some shortening and focussing.
- Should the abbreviation for "Soft tissue tumor surgery complexity score" not be STS-SCS rather than SCS-STS?
- You should explain the term "whoops" procedure, which is not known to everybody.
- What does a "simulator" and "blood-based solid tumor" mean?
- On which rationale do you assign a lower score to a metastasis than to a G1 tumor?
- How are a "resected structure" and an "involved discipline" defined when calculating the score? This seems very arbitrary. How would you for example count, if a sarcoma surgeon is a trained vascular surgeon and perfomed vascular reconstruction himself instead of involving an additional vascular surgeon?
Reviewer 2 Report
The topic is interesting, however, the paper is disorganized and must be deeply improved.
Introduction is too long.
I have major concerns on how the score was built and how Score parameters were chosen
Age: how did you define cut offs? Usually, 65 years cut off is defined as older patients have a reduced administration of Chemotherapy (Acem et al European Journal of Cancer 2020)
What does "simulator" stand for?
Histology and grade must be considered separately. Infiltrative and non-infiltrative subtypes need to be considered in the score.
Vessels involvement must be considered (Sambri et al Cancers 2021)
Type of reconstruction is very heterogeneous
References are very few. Please add related references in order to improve the score as suggested.